# Interactions between Dietary Micronutrients, Composition of the Microbiome and Efficacy of Immunotherapy in Cancer Patients

**DOI:** 10.3390/cancers14225577

**Published:** 2022-11-14

**Authors:** Małgorzata Frąk, Anna Grenda, Paweł Krawczyk, Janusz Milanowski, Ewa Kalinka

**Affiliations:** 1Chair and Department of Pneumonology, Oncology and Allergology, Medical University of Lublin, 20-059 Lublin, Poland; 2Department of Oncology, Polish Mother’s Memorial Hospital—Research Institute, 93-338 Lodz, Poland

**Keywords:** microbiome, micronutrients, immunotherapy, cancer

## Abstract

**Simple Summary:**

Immunotherapy is a systemic therapy significant for numerous types of cancer. The search for factors which may improve the effectiveness of the therapy is still ongoing. The correlation between host microbiome and the efficacy of immunotherapy has been confirmed. Nutrients modulate the composition of the microbiome, which can be used to improve treatment. The paper presents the impact of probiotics, prebiotics and micronutrients on particular species of bacteria associated with a significant increase in response to anti-PD1, anti-PD-L1 and anti-CTLA4 immunotherapy. We also present our own investigation on the relationship between the gut microbiome and the effectiveness of immunotherapy in non-small-cell lung cancer patients.

**Abstract:**

The effectiveness of immunotherapy in cancer patients depends on the activity of the host’s immune system. The intestinal microbiome is a proven immune system modulator, which plays an important role in the development of many cancers and may affect the effectiveness of anti-cancer therapy. The richness of certain bacteria in the gut microbiome (e.g., *Bifidobacterium* spp., *Akkermanisa muciniphila* and *Enterococcus hire*) improves anti-tumor specific immunity and the response to anti-PD-1 or anti-PD-L1 immunotherapy by activating antigen-presenting cells and cytotoxic T cells within the tumor. Moreover, micronutrients affect directly the activities of the immune system or regulate their function by influencing the composition of the microbiome. Therefore, micronutrients can significantly influence the effectiveness of immunotherapy and the development of immunorelated adverse events. In this review, we describe the relationship between the supply of microelements and the abundance of various bacteria in the intestinal microbiome and the effectiveness of immunotherapy in cancer patients. We also point to the function of the immune system in the case of shifts in the composition of the microbiome and disturbances in the supply of microelements. This may in the future become a therapeutic target supporting the effects of immunotherapy in cancer patients.

## 1. Introduction

The health of the human body consists of a series of processes and reactions occurring constantly in all its tissues. Perfect balance is the guarantee of a long and good quality life. Immunity is vital in protecting against various damaging factors from the external environment as well as from the inside of the body. The human microbiome is a well-known component of host immunity. It includes not only bacteria but also other microorganisms such as fungi, archaea and protozoa [1]. The microbiome influences numerous essential functions of the body. Human health and balance depend on its qualitative and quantitative composition. Its composition is unique to the individual and depends on the diet, lifestyle and medications taken. The microbiota is the body’s first line of defense. It mediates the host’s interactions with the external environment (including food, environmental toxins, bacteria, viruses, parasites and fungi) on the skin and mucous membranes in the intestines, lungs, vagina and even the cornea [2]. The digestive tract, especially the colon, has the largest concentration of microorganisms—the number of their cells is 10 times greater than the number of all host cells (1 × 10^14^) weighing about 1.8 kg. The number of their genomes is a hundred times greater. In the human body, there are on average about 160 species of bacteria species [1,3]. The density of the microbiome (measured as colony-forming units (CFUs) per 1 mL) increases from the duodenum (≃10^1–3^ CFU/mL) to the ileocecal valve (≃10^10^ CFU/mL) and reaches the highest concentration in the colon (≃10^11–12^ CFU/mL) [4]. Most of the human bacteria are of the following types: *Firmicutes*, *Bacteroidetes*, *Actinobacteria* and *Proteobacteria*, and strictly anaerobic: *Bacteroides*, *Fusobacteria*, *Proteobacteria*, *Eubacteria*, *Bifidobacteria*, *Clostridia*, *Vermcomicrobia*, *Cyanobacteria*, *Spirochaeates*, *Peptostreocptocci*, and *Ruminococci*. The dominant types are *Firmicutes* and *Bacteroidetes* [4,5,6]. The consistency of the microbiome depends on the diet—for example, *Bacteroides* more often correlates with the high content of fats and animal proteins in the food, and the prevalence of *Prevotella* is associated with the high content of simple sugars [7].

The microbiome in the first place ensures the maintenance of the correct structure of the intestinal mucosa and protects against pathogens. It plays an active part in the fermentation of nutrients that human enzymes cannot digest. Furthermore, it processes endogenous compounds produced by microorganisms and the host itself, providing unique metabolites necessary for proper functioning, such as lipopolysaccharides (LPS), short-chain fatty acids (SCFAs) and tryptophan. It takes part in the synthesis of vitamins, among others K and B vitamins, mainly riboflavin, niacin, biotin, folic acid, pyroxidine, and cobalamin [8]. It breaks down mutagenic carcinogens (heterocyclic amines and N-nitroso compounds) deriving from a diet rich in red meat. In addition, intestinal microorganisms are involved in the synthesis of amino acids, including lysine and threonine. Fermentation by the intestinal microbiome provides up to 10% of energy from food [9]. Therefore, it may be involved in the regulation of body weight and the amount of adipose tissue present in the system [10]. It is believed that a rich, varied microbiome is most desirable, and a depleted microbiome promotes disease. A relationship between disturbances in the microbiome and the occurrence of at least 25 diseases has been shown [11].

Microbiome disorders are common in gastrointestinal diseases—inflammatory bowel disease (IBD), Crohn’s disease, ulcerative colitis, food intolerance, digestive disorders, irritable bowel syndrome (IBS) and colorectal cancer [12,13]. Interestingly, the influence of microbiome extends well beyond the intestinal boundaries and affects not only intestinal homeostasis but also the entire organism. Its validity is observed in metabolic diseases such as non-alcoholic steatohepatitis (NASH), diabetes, dyslipidemia, glucose intolerance, insulin resistance, obesity, hypertension, bone density disorders, allergic diseases and autoimmune diseases such as rheumatoid arthritis and multiple sclerosis [11,14,15]. Also, the correlation with disorders of the nervous system such as Alzheimer’s disease, autism spectrum disorders, depression, chronic fatigue syndrome and Parkinson’s disease has also been shown [9,16]. The subject of this paper, dysbiosis of the intestinal microbiome, mediates the development of numerous neoplastic diseases and influences the effects of oncological treatment [17,18].

By correlating the above information with the scientific reports of the last decade, it should be understood that humans and their microbiota do not exist in isolation, independently of each other, but form a complex meta-organism called “holobiont”. The microbiome regulates itself as well. The huge number of microorganisms requires the implementation of control mechanisms by which the host assesses the state of colonization and reacts to deviations from homeostasis [19,20]. Identifying the mechanisms of interaction between the organism and the gut microbiome is a major challenge for public health in developing new preventive or therapeutic strategies. Owing to achievements in the fields of genetics, molecular biology and bioinformatics, it has become possible to study the microbiome in more detail.

## 2. The Immunomodulatory Activity of the Intestinal Microbiome

Microorganisms participate in the host’s immune system maturation process. They contribute to the formation of innate and adaptive immunity on many levels. In experiments with mice, it was shown that germ-free (GF) individuals devoid of gut microbiota had severe immunodeficiencies. Inter alia, absence of mucosa, altered IgA secretion, reduced size and functionality of Peyer’s patches and mesenteric lymph nodes were observed [21]. Studies at the molecular matter have shown that the differentiation of T cells into T regulatory cells (Tregs) and their functional maturation do not take place in the thymus but in the intestine, with the participation of the commensal microbiome [22].

From the immunological point of view, microbes are acknowledged as pathogens by the host’s immune system which recognizes and eliminates them. The relationship with the microbiome is different—the immune system has evolved to coexist with microbes in symbiosis. These bilateral interactions enable tolerance of commensal bacteria and food antigens, and, at the same time, intestinal bacteria enable the recognition and destruction of opportunistic bacteria, thus preventing bacterial invasion and infection [23]. The intestines contain gut-associated lymphoid tissue (GALT) organized into lymphoid follicles (Peyer’s patches) and containing antigen-presenting cells (APCs), innate lymphoid cells (ILCs), CD4+ and CD8+ T cells and B cells, and many other immune cells. Communication between the GALT system and the intestinal lumen is provided by the intestinal epithelium (IEC) rich in intraepithelial lymphocytes (IELs) and Paneth cells that secrete antimicrobial peptides. In the intestinal epithelium, there are microfold cells (M cells). M cells have the unique ability to take up antigens from the small intestine lumen via endocytosis, phagocytosis or transcytosis. These antigens are delivered to dendritic cells (DCs) and other APCs [23].

The pathogen-related molecular patterns (PAMPs), which are recognized by the pattern recognition receptors of immune cells, mainly Toll-like receptors (TLRs), are of great importance in immunological communication and distinguishing one’s own cells from pathogens [24]. It has been proven that the activation of TLRs led to an increased synthesis of peptides against microorganisms: diptericins (against Gram-negative bacteria), defensins (against Gram-positive bacteria) and drosomycins (antifungal). So far, 10 types of TLR have been identified in humans. They are located in the intestinal epithelial wall: on dendritic cells, macrophages, mast cells, natural killer cells (NK), eosinophils, and on T (including Treg cells) and B cells, as well as in epithelial cells, vascular endothelium, fibroblasts, cardiomyocytes, keratinocytes and adipocytes. The immune response through Toll-like receptors is part of the non-specific (innate) response. PAMPs include mainly lipopolysaccharide, flagellin (a protein of cilia from Gram-negative bacteria) and peptidoglycan, as well as lipoproteins, bacterial polysaccharides, lipoteichoic acid, unmethylated CpG sequences and bacterial wall proteins derived from their breakdown (e.g., under the influence of an antibiotic). TLR2, TLR3, TLR4 and TLR5 are of the greatest importance in PAMP-mediated activation of the immune system [25]. TLR2 ligands are bacterial lipoproteins, peptidoglycan, lipoteichoic acid, zymosan, glycolipids, bacterial porins and lipoarabinomannan. The ligand for the TLR4 receptor is lipopolysaccharide, while the TLR5 receptor recognizes flagellin. The TLR3 receptor is involved in the recognition of microbial nucleic acids—double strand RNA (dsRNA) and synthetic polyinosinic polycytidylic acid (poly I:C). Activation of TLRs in antigen-presenting cells enhances the processes that result in the induction of a specific response [26,27,28].

Upon activation by an antigen, dendritic cells migrate to the mesentery lymph nodes of the small intestine and colon, where they stimulate the conversion of naïve T cells to CD4+ T lymphocytes. Newly formed regulatory CD4+ T and Th17 cells are particularly important due to their intestinal tropism [23]. Tregs return to the gut where they induce immune tolerance directly as well as through the production of immunosuppressive cytokines such as interleukin 10 (IL-10), TGF-β (tumor growth factor) and IL-35 to prevent autoimmunity [4]. Moreover, some bacteria have the ability to maturate Tregs and stimulate TGF-β via an alternative pathway involving polysaccharide A (PSA) and TLR2 receptors on dendritic cells [23]. Live bacteria can also induce an IgA plasma cell response [29].

Th17 cells are located in the *lamina propria* of the mucosa of the small and large intestines and protect the body against bacterial and fungal infections. By producing cytokines, they stimulate the intestinal epithelium to produce tight junctions that seal the intestinal barrier and produce proteins against pathogens. Th17 cells secrete, inter alia, IL-17, which stimulates epithelial and endothelial cells, fibroblasts and macrophages to produce other cytokines, such as IL-6, IL-8 and TNF (tumor necrosis factor). Moreover, it increases secretion of granulocyte and macrophage colony stimulating factor (GM-CSF) and stimulates the maturation of DCs. When activated by dendritic cells, B and T cells (including Tregs and Th17) have the ability to migrate throughout the body wherever they are needed [30]. It is worth noting that migrating Th17 cells have significant plasticity of action, altering cytokine production depending on the presence or absence of inflammation. Thanks to the cascade of reactions taking place with the participation of the microbiome, the body is able to trigger a strong immune reaction even in distant places.

Therefore, the gut microbiome contributes to the establishment of a proper Th1/Th2 balance. Dysbiosis stimulates the prevalence of Th2, which may manifest in allergic disorders. In germ-free mice, administration of *Bacteroides fragilis* has been observed to correct T cells deficiency and PSA-mediated Th1/Th2 imbalance [30]*. Bernesiella intestinihominis* stimulates cytotoxic T cells (CTLs). *Bernesiella intestinihominis* and *Bilophila* increase the pro-inflammatory Th1 response. *Escherichia coli* and *Escherichia coli Nissle* (EcN) increase the production of cytokines such as IL-6, IL-8 and IL-1β via TLR [31]. *Enterococcus hirae* enhances the Th17 cells response and may increase the ratio of CTLs and Tregs [30]. Some intestinal bacteria have the ability to produce anti-inflammatory cytokines, including IL-10, IL-25, IL-33, TGF-β, and thymic stromal lymphopoietins. These include *Bacteroides*, *Lactobacillus acidophilus*, *Lactobacillus murinus*, *Lactobacillus reuteri*, *Helicobacter hepaticus*, *Faecalibacterium* and some strains of *Clostridia* [22,30]. Particularly noteworthy is *Bacteroides thetaiotaomicron*, which has a significant role in the production of defensins. These bacteria stimulate the intestinal epithelium to produce α- and β-defensins, antimicrobial peptides (AMP), type C lectins and hydrolytic enzymes. They also promote the expression of matrix metalloproteinases (MMPs), which are necessary for the activation of defensins. *B. thetaiotaomicron* has the ability to disrupt NFκB activation in the peroxisome proliferator-activated γ-receptor (PPARγ) pathway and thus reduce the inflammatory response of the immune system. In the absence of danger, it is possible to maintain a non-inflammatory state [7].

## 3. The Role of Dietary Fiber and Short-Chain Fatty Acids

Short-chain fatty acids (SCFAs) are products of intestinal bacterial metabolism, resulting from the fermentation of dietary fiber [32]. Dietary fiber (DF) are edible carbohydrate polymers with at least three monomeric units, either derived from natural sources such as cereals, legumes, fruits and vegetables or obtained from food raw materials by physical, enzymatic, chemical or synthetic methods [33]. These polymers are resistant to endogenous digestive enzymes in the human small intestine and fermented by bacteria in the large intestine. Soluble fiber is highly fermented, and it is the main source of energy for the microbiome and thus causes the formation of metabolites such as short-chain fatty acids [34]. Soluble fiber includes pectins, inulin, β-glucans, arabinoxylans, oligosaccharides and guar gum. Insoluble fiber (lignin, cellulose, hemicellulose) is poorly fermented by bacteria due to its inability to retain water. Fermentation is possible in the presence of species and strains with the enzymatic ability to metabolize fiber [33]*. Bifidobacterium, Bacteroides* and *Prevotella* have a major part in this process. DF fermentation is associated with a strong anti-inflammatory effect, lowering the pH of the colon, increasing the bacterial diversity of the large intestine, reducing pathogens, stimulating the production of antioxidant compounds and vitamins, as well as regulating the epithelial barrier.

SCFAs are organic compounds consisting mainly of acetate, propionate and butyrate. In addition to lowering intestinal pH, they increase the bioavailability of some metals, e.g., iron [35]. *Bacteroides thetaiotaomicron*, responsible for the production of acetate and propionate, promotes the differentiation of goblet cells in the intestine and the expression of mucin-related genes. *Faecalibacterium prausnitzii,* by consuming acetate reduces its influence on mucus and prevents its overproduction. *F. prausnitzii* is responsible for the production of butyrate, thanks to which it maintains the appropriate structure and composition of the intestinal epithelium [36]. 

SCFAs lead to GPR43-dependent stimulation of Tregs, and also by induction of histone H3 acetylation [37,38]. This ensures a balance between anti-inflammatory Tregs and pro-inflammatory Th17 cells [39]. They increase the concentration of IL-10 which is produced after the recognition of polysaccharide A by plasma dendritic cells and reduces inflammation [40]. Microbial SCFAs and dietary fiber fermentation products can stimulate the population of myeloid DCs in the bone marrow and stimulate the phagocytic capacity of these cells. SCFAs also show anti-inflammatory effects by inhibiting NF-κB, binding to G protein-coupled receptors 43 and 41 (GPR43 and GPR41) [31]. Higher amount of SCFAs inhibit the expression of proinflammatory tumor necrosis factor in mononuclear cells and neutrophils and may lead to inactivation of NF-κB. This interaction promotes signal transduction and alleviates the effects of hypoxia, increases intestinal integrity and prevents LPS translocation to combat inflammation. Additionally, SCFAs increase the activation of histone acetyltransferase and inhibit the histone deacetylase, which influence the development of anti-inflammatory phenotypes in the intestinal microbiome [41]. Moreover, they are involved in epigenetic regulation of inflammation through free fatty acid receptors (FFARs) [31,42]. The most important SCFA-producing bacteria are *Faecalibacterium prausnitzii, Eubacterium, Roseburia, Bifidobacterium longum, Ruminococcus, Alistipes* and *Lactobacillus*. Polysaccharide A is produced by *Bacteroidales, Erysipelotrichales, Clostridiales* and *Bacillales* [39].

Numerous studies reflecting the effects of fiber on the microbiota have been conducted. The use of β-glucan in the diet after 4 weeks resulted in an increase in the generalized diversity of the microbiome and an increase in the total amount of bacteria as well as the richness of *Bifidobacterium* spp. and *Akkermansia municiphila* compared to the control group without β-glucan in the diet [43]. The inclusion of 6.3 g of fiber per day and 2.9 g of β-glucan per day for 6 weeks resulted in an increase in the total amount of intestinal bacteria and a significant increase in *Bifidobacterium* spp. and *Lactobacillus* spp. compared to the control group. The use of a diet containing 3 g of fiber per day excluding β-glucan, however, resulted in a significant decrease in the amount of the above-mentioned bacteria [44]. A study evaluating the effect of β-glucan at a dose of 6 g per day showed a significant increase in microbiological diversity and richness after 4 weeks and an increase in the number of *Bifidobacterium* spp. and *Akkermansia municiphila* compared to the control group without β-glucan in the diet [43]. An increased amount of propionate and a decreased amount of acetate in the faeces has also been shown. The application of a diet with the content of 16 g of fiber/1000 kcal showed a reduction in the contents of *Enterobacteriaceae* after 6 weeks and the growth of *Lachnospira* and *Roseburia* compared to the application of a diet of 8 g of fiber/1000 kcal [45]. Immediate reaction in the composition of the intestinal microbiome after diet modification was proven in KovatchevDatchary’s study. In only 3 days of a high-fiber diet (37.5 g/day), an increase in the amount of *Bacteroidetes* and in the *Prevotella/Bacteroides* ratio was observed [46]. A high-fiber diet leads to the growth of SCFA-producing bacteria, including *Lachnospira, Akkermansia, Bifidobacterium, Lactobacillus, Ruminococcus, Roseburia, Clostridium, Faecalibacterium* and *Dorea* [45,47,48].

## 4. The Effectiveness of Cancer Immunotherapy Depends on the Microbiome

Immunotherapy is widely used in systemic treatment of many types of cancers. In the late nineteenth century, American surgeon William Coley observed spontaneous remission of a malignant sarcoma in a patient with comorbid bacterial infection (erysipelas). Inspired by that experiment, Coley started to infect cancer patients with *Staphyloccocus aureus* which causes erysipelas, or by injecting bacterial culture supernatants. In his studies, he achieved remission of various types of cancers. This became the basis of nonspecific active immunotherapy, used, inter alia, in BCG vaccine (Bacillus Calmette-Guérin) for intravesical infusion in patients with surgically removed bladder cancer.

Immunotherapy is based on the immune potential of the host. Under regular conditions the human immune system destroys cancer cells that arise spontaneously in the body. To avoid this, cancer cells have developed mechanisms to escape from the surveillance of the immune system, thus ensuring their survival and further growth. This phenomenon was discovered in 1967 by Burnet and Thomas and ushered in a new era of cancer treatment. The avoiding process is various and multi-stage. First, cancer cells tend to be invisible by reducing or completely removing the expression of major histocompatibility complex class I molecules (MHC), their antigens and co-stimulatory molecules. The expression of proteins related to the recognition and transport of antigens across the cell membrane is inhibited. As a result, tumor cells are not recognized by cytotoxic T cells. The second mechanism of cancer cells’ defense is the production of anti-inflammatory factors such as prostaglandins, histamine, epinephrine, arginase, TGF-β and IL-1 [6]. An “immune desert” is formed, poor in non-specific (NK cells, macrophages, neutrophils) and specific (effector cytotoxic T cells, Tregs, T helper cells) immune cells as well as pro-inflammatory cytokines (e.g., IL-2, IL-12, TNF-α). A non-immunogenic “cold tumor” is characterized by a low response to immunotherapy. The third mechanism by which a tumor escapes from immune surveillance is the most important in the context of immunotherapy. It consists in the direct neutralization of immune cells by interaction of immune checkpoints such as cytotoxic T cell antigen 4 (CTLA-4) and programmed death 1 (PD-1, CD279). PD-1 is located on the surface of macrophages, monocytes as well as T and B cells. It performs a major role in suppressing the immune system. Tumor cells develop ligands for PD-1 (PD-L1 and PD-L2) that bind the PD-1 receptor on T cells and inhibit their activity. CTLA-4 is located on lymphocytes, and its displacement of CD28 (costimulatory molecule) from binding to CD80 (B7-1) and CD86 (B7-2) on APCs causes T cells anergy [5]. To defeat these mechanisms is a challenge for immunotherapy research. Its action is based on the use of anti-CTLA-4, anti-PD-1 or anti-PD-L1 antibodies, which, after binding to immune checkpoints, lead to the activation of lymphocytes. Tumors become rich in immune cells and pro-inflammatory cytokines such as TNF-α, which is a prerequisite for the effectiveness of immunotherapy. This ensures the destruction of the tumor not by a cytotoxic drug but by the host’s own immune cells, so it is potentially possible to cure the tumor even at a very advanced stage.

Immune cells’ responses depend on many factors. One of them is the gut microbiome [30,31]. The first scientific reports indicated an increased incidence of colorectal cancer in patients with intestinal dysbiosis, which is beyond doubt. Chronic inflammation, IBD, food intolerance, infections, colonization with pathogenic strains, butyrate deficiency, a high-protein diet and products of its metabolism leading to a decrease in the pH in the colon, all create favorable conditions for the intensification of the inflammatory process and carcinogenesis. 

However, in the development of clinical trials, a correlation was observed between the effectiveness of immunotherapy (but also chemotherapy and radiotherapy) for various types of cancer located outside the large intestine (e.g., in the lung) and the composition of the intestinal microbiome. Initial observations were made in mice with solid tumors receiving broad-spectrum antibiotic therapy. A significantly worse response to the immunotherapeutic PD-1 blockade has been demonstrated. A similar dependence took place in a group of mice grown in germ-free conditions. The reason was the significantly low level of TNF-α and T cells activation. Oral supplementation with intestinal bacteria strains restored the ability to respond to immunotherapy [49]. High levels of TNF-α correlated with a good response to immunotherapy and increased levels of *Alistipes shahii* in mice gut [50,51]. The enrichment of the microbiome of mice with bacteria of the *Bifidobacterium* species resulted in an increase in the level of T CD8+ cells in the tumor microenvironment and a delay in tumor growth. The combination of *Bifidobacterium* supplementation with anti-PD-L1 antibody resulted in almost complete tumor remission in mice [50]. An analogous response was observed for anti-CTLA-4 antibody in mice with melanoma, sarcoma and colorectal cancer. *Bacteroides thetaiotamicron* and *Bacteriodes fragilis* were necessary for the efficacy of the anti-CTLA4 antibody. These bacteria induced interleukin-12-dependent antitumor T-helper 1 lymphocyte responses [52].

In the group of patients not treated with antibiotics or with short-term exposure to antibiotics (<7 days), longer overall survival (OS) and progression-free survival (PFS) were reported, compared to patients with longer exposure [53]. The effect of antibiotic administration was strongest within 30 days prior to treatment with checkpoint blockade [54]. Of 60 patients with advanced cancer (including lung, kidney, melanoma, hepatocellular carcinoma, head and neck cancer and urothelial carcinoma) treated with anti-PD-1 antibodies, 17 patients were receiving antibiotics within 2 weeks before or after starting treatment for various bacterial infections. These patients had a worse response to immunotherapy. Treatment with broad-spectrum antibiotics (against Gram-positive and Gram-negative bacteria, both aerobic and anaerobic) resulted in shorter overall survival compared to patients treated with narrow-spectrum antibiotics (Gram-positive bacteria only) [55,56]. Heumer et al. conducted a study of 30 patients with non-small-cell lung cancer (NSCLC) of whom 11 received an antibiotic within a month before or after starting treatment with immune-checkpoint inhibitors (ICIs). Authors observed a significantly shorter PFS and OS in patients treated with antibiotics (2.9 and 7.5 months) versus patients without antibiotics (13.1 and 15.1 months) (*p* = 0.028 and *p* = 0.026) [57]. The above observations are justified by a decrease in the diversity and richness of the gut microbiome, which leads to a reduction in the production of pro-inflammatory cytokines and less pronounced tumor necrosis.

## 5. Gut Microbiota Species Associated with the Efficacy of Immunotherapy

The influence of the intestinal microbiota on the efficacy of ICIs has been noted through the remote modulation of lymphoid and myeloid cells. The microbiome triggers the activation mechanism of IL-12 dependent Th1 cells with cross-reactivity to tumor and bacterial antigens and stimulates DCs [7]. Mucin-eating bacteria (i.a. *Bifidobacterium longum* or *Akkermansia muciniphila*) have been linked to a better response to treatment [58]. It was confirmed that *Bacteroides* (*Bacteroides thetaiotaomicron* and *Bacteroides fragilis*) increases the effectiveness of anti-CTLA-4 therapy and restores its effects after administration of antibiotics [59,60]. Patients whose gut microbiome was rich in *Bacteroides* had increased Th1 cells and decreased Tregs cells and myeloid suppressor cells (MDSC). In mice with large numbers *of Bacteroides* (mainly *B. fragilis*) and *Burkholderia* in the intestines, slower growth of various tumors was observed [7]. In mice, a reduction in MCA205 sarcoma was observed during anti-CTLA-4 therapy when *Bacteroides fragilis, Bacteroides thetaiotamicron* and *Burkholderia* species were present in their organisms. These bacterial species may influence the IL-12 dependent Th1 immune response, which enables better disease control [52]. However, it is believed that other common species of *Bacteroides* do not have an impact on the efficacy of anti-CTLA-4 treatment [61].

Patients with melanoma treated with anti-CTLA-4, and with high *Faecalibacterium* abundance and low abundance of *Bacteroidales*, had significantly prolonged PFS compared to those with low and high abundance of these bacteria (*p* = 0.03 and *p* = 0.05) [62,63]. However, the presence of more *Bacteroidetes*, including *Bacteroidaceae*, *Rikenellaceae* and *Barnesiellaceae*, had a preventive effect against the onset of complications of immunotherapy in the form of autoimmune colitis after immunotherapy [63,64].

The presence of *Bifidobacterium* spp. in the gut microbiome has been associated with greater anti-PD-L1 treatment efficacy in melanoma patients. Oral administration of probiotics containing *Bifidobacterium* to mice increased the anti-tumor efficacy of PD-L1 blockade and led to an almost complete inhibition of tumor growth [50]. Gopalakrishnan et al. showed that in melanoma patients, a higher level of *Faecalibacterium prausnitzii, Ruminococcus bromii, Enterococcus faecium, Collinsella aerofaciens, Bifidobacterium adolescentis, Klebsiella pneumoniae, Veillonella parvula, Parabac-teroides merdae* and *Lactococcus formicus* correlated with a better response to anti-PD-1 immunotherapy [49,61,62,65]. In metastatic melanoma patients, the richness of *Bifidobacterium longum, Bifidobacterium adolescentis, Enterococcus faecium, Collinsella aerofaciens, Lactobacillus species, Klebsiella pneumoniae, Veillonella parvula* and *Parabacteroides merdae* was higher among responders to anti-PD-L1 treatment, whereas *Roseburia intestinalis* and *Ruminococcus obeum* were significantly higher among the non-responders [65].

More detailed studies were carried out in a group of patients treated with anti-PD-1 with various types of cancer (NSCLC, melanoma, renal cell carcinoma, etc.). Multiparameter immunohistochemistry (IHC) confirmed a higher density of CD8+ T cells in the tumor environment in patients with a better response than in patients with a worse response to immunotherapy (*p* = 0.04). A high level of T CD8+ in tumor tissue correlated with an increased abundance of the genus *Faecalibacterium* of the *Ruminococcaceae* family and of the order *Clostridiales.* Flow cytometric analysis showed that patients with high levels of *Clostridiales, Ruminococcaceae* or *Faecalibacterium* in the gut had higher levels of CD4+ and CD8+ effector T cells in the peripheral blood. These patients responded more frequently to anti-PD-1 therapy. However, the patients with high abundance of *Bacteroidales* in gut microbiome had higher levels of regulatory Tregs and MDSC in the peripheral blood and they were less responsive to ICIs treatment [66,67]. Higher content of *Faecalibacterium* in the gut microbiome was connected with a high density of immune cells and high concentration of pro-inflammatory and antigen-presenting cytokines compared to patients with a large amount of *Bacteroidales* [68].

In other research, Tanoue et al. indicate 11 commensal strains which induce CD8+ T cells and interferon-γ in the intestine of germ-free mice and cause significant increase of the response to anti-PD-1 or anti-CTLA-4 antibodies [69]. These strains are: *Ruthenibacterium lactatiformans, Bacteroides uniformis, Bacteroides dorei, Fusobacterium ulcerans, Eubacterium limosum, Phascolarctobacterium succinatutens, Paraprevotella xylaniphila, Parabacteroides distasonis* and *Parabacteroides johnsonii*. Bacteria may increase the efficacy of checkpoint inhibitors therapy by stimulating dendritic cells to secrete IL-12 and stimulate differentiation of cytotoxic T cells [56,61].

Song et al. analyzed the gut microbiome in NSCLC patients treated with ICIs. PFS ≥ 6 months was associated with significantly increased diversity in the gut microbiome compared to the PFS < 6 months. In the PFS ≥ 6 months group, a richness of *Methanobrevibacter* and *Parabacteroides* was observed, while in the PFS group < 6 months an enrichment in *Selenomonadales, Negativicutes* and *Veillonella* was noted [70]. Jin et al. showed a relationship between high abundance of *Alistipes putredinis, Prevotella copri* and *Bifidobacterium longum* and greater efficacy of anti-PD-1 treatment in NSCLC patients. Multicolor flow cytometry showed that patients with a greater variety of gut microbes had a higher percentage of peripheral blood NK cells and memory CD8+ T cells [71]. Valuable research was also presented by Lee et al., who demonstrated the richness of the microbiome in *Bifidobacterium bifidum* in a group of NSCLC patients with a better response to anti-PD-L1 treatment [72]. The presence of *Bifidobacterium spp*. is associated with a response due to an increase in CD8+ T cells and DCs in the tumor microenvironment. Consequently, a higher level of IFN-γ is produced [62]. *Bifidobacterium spp*. improves tumor-specific immunity and the response to anti-PD-L1 immunotherapy by activating antigen-presenting cells inside the tumor [50].

*Akkermansia muciniphila* and *Alistipes indistinctus* have a marked effect on the anti-PD-1 response in NSCLC and renal cell carcinoma (RCC). Their disproportionate number was recorded in the stool of people with a better response to treatment (partial response or stable disease) compared to those without response (progression). High abundance of *Akkermanasia muciniphila* was significantly associated with response to ICIs therapy in NSCLC and RCC patients (*p* = 0.004). *A. muciniphila* was also more frequent in the stool of patients with PFS longer than 3 months (*p* = 0.028) [49]. Oral supplementation with specific *Akkermansia muciniphila* may restore the responsive phenotype in non-responders [49,73]. A high proportion of the following species among microbiota increases the effectiveness of anti-PD-1 immunotherapy in NSCLC patients: *Akkermansia muciniphila, Alistipes indistinctus, Bifidobacterium breve, Propionibacterium acnes, Prevotella copri, Rikenellaceae, Staphylococcus aureus, Streptocptoccus prausnitzi, Bacteroides plebeius, Enterococcus hirae* and *Enterobacteriaceae* [Table 1]. In contrast, in patients with a good response to anti-PD-1 therapy, *Ruminococcus bromii, Dialister and Sutterella* occurred less [7,71,73]. Moreover, abundance of *Ruminococcus* unclassified was detected in the gut microbiome of patients with a poor response [71].

## 6. Our Own Observations on the Relationship between the Gut Microbiome and the Effectiveness of Anti-PD-1 or Anti-PD-L1 Immunotherapy in NSCLC Patients

The study of the microbiome is possible thanks to next-generation sequencing (NGS), which allows the identification of the composition of the broad range of microbiota without the need for culture. The bacterial hypervariable region of 16S RNA genes, which are characteristic of individual bacteria, is analyzed in this approach. In our preliminary study of the gut microbiome of patients with advanced non-small-cell lung cancer treated with anti-PD-1 or anti-PD-L1 antibodies, we found that high abundance of the *Akkermansiaceae* family (of which *Akkermansia mucinifila* is a major member) is a favorable predictor of immunotherapy efficacy (PFS prolongation), thus confirming the results in the literature [76]. Nevertheless, an excessively high percentage of *Akkermansiaceae* may be an indicator of dysbiosis and low diversity of the gut microbiome, occurring, for example, as a result of the use of antibiotics (especially broad-spectrum antibiotics). This is a very important factor, due to the fact that a high diversity in the composition of the microbiome is associated with stimulation of the immune system in oncological diseases, and with an increase in the efficacy of immunotherapy.

In 37 NSCLC patients not treated previously with antibiotics, the abundance of *Oscillospirales* (especially *Oscillospirales UCG-010*) was significantly positively correlated with PFS duration. Moreover, high content of *Staphylococcaceae* and *Leuconostocaeae* was associated with a significantly shorter PFS (data not published). In the whole group of patients treated and not treated with antibiotics (n = 47), we observed even more significant correlations between PFS and the abundance of the mentioned bacterial groups. In addition, we noticed that in the whole group of patients a high abundance of *Veillonellaceae* was correlated with shorter PFS, which seems to be in line with the results of Song et al. [70]. Moreover, Cekikij et al. indicated that patients with CRC non-resistant to ICIs therapy had an increased content of *Akkermansia*, *Clostridium sensu stricto 1* and *Oscillospiraceae* and decreased content of *Streptococcus* and *Leuconostoc* [77]. Due to the fact that PFS is not the best indicator of the effectiveness of immunotherapy, our 47 advanced NSCLC patients with microbiome genotyping are under surveillance in order to obtain prospective long-term observations in terms of overall survival.

## 7. Probiotics and Prebiotics

Knowing the influence of particular species of bacteria on the immune system and the effectiveness of cancer immunotherapy, it is worth considering how we can interfere with the quantitative and qualitative composition of the microbiome to a patient’s benefit. The experiences to date confirm that even a few days’ diet modification has a clear impact on the microbiome [46]. Modifying the microbiome through the diet can be directly through the supply of live bacteria (probiotics), which are a natural component of food. In this way, we provide the organism with fermenting bacteria, such *as Lactobacillus, Bifodobacterium, Enterococcus* or *Pediococcus* [7,78,79]. Dairy products, mainly yoghurts, kefirs and cheeses, are commonly enriched with *Bifidobacterium longum, B. lactis, Lactobacillus acidophilus, L. casei* and *L. paracasei* [80]. Buttermilk provides *L. lactis* and *L. bulgaricus*. Fermented vegetables such as pickled cucumbers or cabbage are rich in *Lactobacillus* species, especially *L. plantarum* and *L. brevis*, while kimchi provides *Lactococcus lactis*. Fermented soybean products such as tempeh and miso contain large amounts of *Enterococcus faecium* and *Lactobacillus* species. Live bacteria can also be supplied in the form of ready-made preparations, standardized in relation to the quantitative and qualitative composition [78]. It should be strongly emphasized that the safety and efficacy of large amounts of probiotics in cancer patients has not been demonstrated. There are also no data regarding whether such a procedure has an impact on the effectiveness of immunotherapy. Probiotics can disrupt the natural composition of the microbiome, which may be counterproductive.

Modification of the microbiome is also possible by supplying the body with ingredients that contribute to its enrichment. Prebiotics are commonly known substrates in the form of non-digestible food ingredients (carbohydrates) which are components of the dietary fiber fraction, primarily oligosaccharides, among which the most important are fructooligosaccharides, lactulose and soy oligosaccharides, and polysaccharides, among which the most important are inulin, β-glucans, and pectin [78]. Foods naturally rich in prebiotics include chicory, onions, asparagus, garlic, potatoes, bananas, soybeans (including soybean whey), artichokes, wheat, oats, barley, Jerusalem artichoke, tomatoes and honey. Fiber selectively stimulates growth and increases the activity of microorganisms that are beneficial to health in the intestinal microbiota, regulates the pH of the intestinal contents, and provides a breeding ground for bacteria, stimulating the production of SCFAs, butyric acid and vitamins. A diet rich in dietary fiber results in an increase in the total amount of intestinal bacteria, increasing the diversity of the microbiome with a significant increase in *Bifidobacterium spp*. and *Lactobacillus* spp. [43]. A high-fiber diet enriches the microbiome with bacteria, which has been associated with increased immunotherapy effectiveness in various cancers, which was widely described above.

## 8. Micronutrients, Gut Microbiome and Cancer 

As opposed to the commonly known beneficial effects of fiber, knowledge about the influence of micronutrients on the development of intestinal microbiota beneficial for humans is less widespread. However, micronutrient deficiencies have been linked to changes of bacterial species in the human gut microbiota affecting the host regulation of immune responses. Moreover, malnutrition is one of the common symptoms of cancer [81]. The supply of selected elements and vitamins could promote the richness of the intestinal microbiome and increase the amount of bacteria that are vital to higher effectiveness of oncological treatment (Table 2).

### 8.1. Vitamin A

In a large study involving 306 children, vitamin A supplementation compared to the placebo resulted in a higher concentration of *Bifidobacterium* in the faeces. These observations were made in boys, but this difference was not present in girls [82]. In girls in late infancy, a positive correlation was found between the concentration of retinol in the plasma with *Bifidobacterium* and *Akkermansia* abundance. In a group of patients with retinol deficiency, a higher concentration of *Enterococcus faecalis* in the faeces was demonstrated [83]. Studies by Mandala et al. showed that an increase in retinol consumption was associated with the increase in *Proteobacteria* to *Actinobacteria* and *Proteobacteria* to *Firmicutes* ratios [84]. However, research results on the influence of vitamin A supplementation on the composition of the microbiome are inconsistent. Another study in a group of 64 children showed that after vitamin A intake, stool samples were dominated by *Bacteroidetes* (46%) as well as *Proteobacteria, Actinobacter, Enterobacter* and *Bifidobacterium* [85].

Consumption of carotenoids in the diet may reduce the risk of colon cancer, and conversely, dietary beta-carotene consumption was inversely related to the incidence of colorectal adenoma. A link has been shown between vitamin A deficiency and an increased incidence of different cancers, e.g., breast, cervix, lung, skin, mouth, prostate and leukemia [86,87]. Vitamin A has been used in clinical trials, alone or in combination with chemotherapy, to treat breast cancer, colorectal cancer, hepatocellular carcinoma, melanoma, neuroblastoma, glioblastoma, skin T-cell lymphoma, lung cancer, prostate cancer, gastric cancer and pancreatic cancer [88]. Interesting results were presented by Pastorino and co-authors. In a group of 307 patients with stage I NSCLC after surgery, the adjuvant effect of high-dose vitamin A was examined. One group of patients received retinol palmitate (orally 300,000 IU daily for 12 months) and the control group received a placebo [89]. After a median follow-up of 46 months, 131 relapses were observed. Recurrence of the disease occurred in 56 patients receiving vitamin A (37%) and in 75 patients in the control arm (48%) [89]. The development of the second primary tumor occurred with a similar frequency in patients receiving and not receiving vitamin A supplementation (29% vs. 33%). Carotenoids in human breast cancer cell lines inhibit cell proliferation and increase cancer cell apoptosis [90,91]. Vitamin A supplementation reduced the liver metastases of colon cancer [92] and reduced melanoma metastases in mice [93]. Despite numerous scientific studies and clinical trials, Vitamin A has not been widely used in the prevention and treatment of cancer. Only all-trans retinoic acid (ATRA) has been registered for the treatment of acute promyelocytic leukemia (APL).

In the context of cancer therapy, vitamin A has a wide range of effects on the immune system. Retinol is required for B cells stimulation, thymocytes growth, myeloid-derived suppressor cells (MDSC) maturation and NK cells activation [94]. Vitamin A is involved in regulating the function of mitochondria and microRNA and it influences tumor stem cells. Retinoids bind to selective proteins in the target cells, such as RAR (retionoic acid receptor), RXR (retinoid X receptor), EGFR (epidermal growth factor receptor), JAK2 and caspase-3 [95]. On the other hand, receptor complexes bind to a selective region of nuclear DNA, which enables the regulation of gene expression and protein synthesis. A total of seventeen gene signaling pathways have been involved in the anti-tumor mechanisms of retinoids [88]. Modulating the appropriate signaling pathways of vitamin A results in inhibition of tumor cells proliferation and tumor growth arrest [88]. There are no scientific reports examining the effect of vitamin A on the effectiveness of cancer immunotherapy, either directly or through the influence of this vitamin on the immune system or microbiome, except for one phase II clinical trial. Five melanoma patients received ipilimumab alone and another five patients ipilimumab plus ATRA. ATRA significantly decreased the frequency of circulating MDSCs compared to ipilimumab treatment alone in advanced-stage melanoma patients. Additionally, ATRA reduced the expression of immunosuppressive genes, including PD-L1, IL-10 and indoleamine 2,3-dioxygenase (IDO), by MDSCs. Furthermore, ATRA did not increase the frequency of grade 3 or 4 adverse events [96].

### 8.2. B Vitamins 

The effect of vitamin B1 (thiamine) was demonstrated on lymphoid tissue—lymph nodes and spleen were smaller in the group of mice that were not given vitamin B1. Upon commencement of supplementation, tissues increased in volume and returned to normal size in 14 days [97]. In studies in mice infected with *Mycobacterium tuberculosis* that were treated with vitamin B1, increased levels of TNF-α and IL-6 in the lungs and enhanced upregulation of CD86 and MHC-II expression on antigen-presenting cells were shown. Further observations indicate that vitamin B1 could regulate the functions of macrophages and regulate the NF-κB signal in macrophages to promote the protective immune response [98]. In a group of 257 patients, the influence of the supply of vitamin B1 in a dose of up to 0.6 mg/1000 kcal/day on the intestinal microbiota was examined. The study showed a significant increase in *Bacteroides, Faecalibacterium* and *Prevotella*. The greatest correlation was found between the *Ruminococcaceae* family and the vitamin B1 supplementation [99].

It has been observed that the microbial pathway of vitamin B2 (riboflavin) synthesis produces metabolites that stimulate the activation of mucosa-bound T cells in intestines and airways, promote a tissue repair response and help to maintain the integrity of the epithelial barrier [100]. Vitamin B2 supplementation resulted in an enrichment of the overall diversity of the microbiome, in particular in *Faecalibacterium* and *Roseburia*, as well as a reduction in the number of *Enterobacteriaceae* [101,102]. The use of high doses of vitamin B2 (100 mg daily) resulted in an increase in *Faecalibacterium prausnitzii* (one of the main producers of butyrate) within 2 weeks of supplementation. A reduction in the number of these bacteria was observed after the end of supplementation. An increase in the number of *Roseburia* species and a decrease in *Escherichia coli* were also shown. Randomized studies of vitamin B2 supplementation at a dose of 75 mg/day showed an increase in *Alistipes shahii* in the gut microbiome [103]. In a study of patients who received vitamin B2 for 2 weeks, there was an increase in *Faecalibacterium prausnitzii* in the faeces, which is a strong producer of SCFAs. Another study, in which supplementation lasted for 3 weeks, showed a reduction in the number of potentially pathogenic *Enterobacteriaceae* (including *Escherichia coli*) [101].

Studies conducted in a group of people supplementing orally with vitamin B3 (nicotinic acid) indicate the effect of this vitamin on the growth of *Bacteroidetes* [103]. To confirm this concept, 10 volunteers received vitamin B3: nicotinic acid (30–300 mg) and nicotinamide (900–3000 mg) to the ileo-colonic region to evaluate direct effects on the gut microbiome. After a period of 6 weeks, a significant increase in the amount of *Bacteroidetes* was observed in the group using nicotinic acid compared to the group using nicotinamide [104]. Vitamin B3 suppresses monocytes by inhibiting the release of inflammatory mediators such as TNF-α, IL-6 and monocyte chemotactic protein-1, and it also reduces the production of pro-inflammatory cytokines by macrophages [105].

Vitamin B12 in food is present in the form of a protein complex and it is released free via pepsin in the stomach. It is absorbed in the small intestine via the intrinsic factor (IF). Vitamin B12 is also produced by intestinal bacteria, which include *Bifidobacterium animalis, B. infantis, B. longum, Lactobacillus plantarum*, *L. coryniformis*, *Bacteroides fragilis*, *Prevotella copri*, *Faecalibacterium prausnitzii* and *Ruminococcus lactaris* [106]. Vitamin B12 produced by *Eubacterium hallii* promotes *Akkermansia muciniphila* growth and propionate production [107]. Additionally, *A. muciniphila* stimulates the growth of *E. hallii* and *Anaerostipes caccae*, which are responsible for the participation in the vitamin B12 biosynthetic pathway. In murine models, vitamin B12 deficiency reduces the number of lymphocytes and CD8+ T cells, increases the ratio of CD4+ cells to CD8+ cells and inhibits the activity of NKT cells. This effect can be reversed with vitamin B12 supplementation. These observations confirm that vitamin B12 contributes to the immune response mediated by CD8+ T cells and NK T cells. Vitamin B12 supplements may reduce the direct toxic side effects of therapy as vitamin B12 is required for DNA synthesis, neural functions and reduction of the severity of drug-induced peripheral neuropathy in patients who receive chemotherapy (especially pemetrexed) [108].

### 8.3. Vitamin C

Vitamin C supplementation significantly increases the biodiversity of the gut microbiome, with a particular abundance of *Collinsella* and fecal SCFA levels, especially butyrate and propionate [103]. In vitro studies showed that high vitamin C concentration causes the growth of *Roseburia, Faecalibacterium, Akkermansia* and *Bifidobacterium*, but this trend has not yet been confirmed in humans. Vitamin C is essential for the survival of commensal anaerobes such as *Bacteroides* [109]. Vitamin C acts as an absorber for radicals (e.g., hydroxyl OH− or superoxide O_2_−), thus generating ascorbyl free radical (AFR) and hydrogen peroxide (H_2_O_2_) or water [94]. Neoplastic cells coexist with increased expression of vitamin C transporters and intensified oxidative processes. In this way, the accumulation of free radicals occurs, which causes an imbalance in the activity of mitochondria and the production of reactive oxygen species. This in turn causes demethylation, damage to DNA strands and biological membranes. In the mechanism outlined above, vitamin C inhibits cancer proliferation and enhances cancer cell apoptosis.

Vitamin C deficiency has been correlated with a higher frequency of gastric cancer and prostate cancer occurrence [94]. Vitamin C stimulates and strengthens the function of leukocytes and neutrophils. Supplementation may enhance the proliferation of T cells and increase the production of cytokines. Numerous studies examine the usefulness of vitamin C as monotherapy or in combination with chemotherapy or immunotherapy in the treatment of many types of cancer [110]. Leukemia, colorectal cancer, melanoma, pancreatic cancer, prostate cancer, NSCLC, breast cancer, ovarian cancer, hepatocellular carcinoma, mesothelioma, thyroid cancer, oral squamous cell carcinoma, neuroblastoma and glioma were the most commonly studied. Studies conducted in animal models have shown inhibition of tumor growth (40–60%) by the use of increased doses of ascorbate (1–4 µg/kg) intravenously or intraperitoneally. The supply of vitamin C was also effective in inhibiting the formation of metastases (50–90%) [110]. In combination with immunotherapy, vitamin C increased the immunogenicity of effector T cells in murine models [111,112]. Complete regression was observed in several mice, and immunity persisted after re-injecting the tumor cells. Influenced by vitamin C, increased tumor infiltration by CD8+ T cells (including cytotoxic T cells) and macrophages was demonstrated, and increased production of IL-12 by antigen-presenting cells was observed. Vitamin C contributes to the transformation of “cold tumors” insensitive to immunotherapy into “hot tumors” susceptible to treatment [110].

### 8.4. Vitamin D

Recent studies show that vitamin D can directly affect the intestinal and respiratory tract microbiome and alleviate dysbiosis [113]. In animal models, C57BL/6 mice reared on a vitamin D-rich diet had 50 times more bacteria in the colon and greater microbial diversity, as well as increased production of SCFAs per gram of dry weight, compared to those fed a low-vitamin D diet [114]. Butyrate increases the expression of the vitamin D receptor (VDR) in intestinal epithelial cells. A similar effect of VDR stimulation occurs after administration of the probiotic strains *Lactobacillus rhamnosus* GG ATCC 53,103 and *L. plantarum* [115]. The intestinal microbiota of mice deficient in VDR shows loss of *Lactobacillus spp*. and an increase in *Proteobacteria spp*. and *Bacteroidetes spp*. [115]. In humans, in a group of 3188 IBD patients, it was observed that higher plasma 25(OH)D3 concentration (27.1 ng/mL) was associated with a significantly reduced risk of *Clostridium difficile* infection [114,116]. In studies of patients with multiple sclerosis (MS), it has been shown that supplementation with 5000 IU of vitamin D3 daily for 90 days increased the number of *Akkermansia* as well as *Faecalibacterium* and *Coprococcus,* which produce butyrate and anti-inflammatory SCFA [117]. A study of MS patients treated with vitamin D3 showed that the supply of this vitamin caused changes in the levels of *Firmicutes, Actinobacteria* and *Proteobacteria* and an increase in the number of *Enterobacteria* compared to those who were not treated with vitamin D3 [118]. The daily intake of 60 µg of cholecalciferol resulted in an enrichment of *Bifidobacterium longum* in stool samples [119]. The use of high doses of 50,000 IU cholecalciferol weekly for 9 weeks reduced the amount of *Bacteroidetes* and *Lactobacillus*, whilst *Firmicutes* and *Bifidobacterium* increased after supplementation [120]. Subsequent studies confirm that high serum vitamin D levels can be correlated with high abundance of several *Firmicutes* such as *Ruminococcus, Coprococcus, Mogibacterium* and *Blautia* [121]. In another study, vitamin D supplementation (300,000 IU over 4 weeks) modified the composition of the gut microbiome resulting in the growth of beneficial bacteria such as *Alistipes, Faecalibacterium, Roseburia* and *Parabacteroides* [122,123]. A correlation was also observed between changes in the composition of the intestinal microbiome and the season of the year. In the summer–autumn period, when sun exposure and serum level of vitamin D are the highest, abundance of *Pediococcus spp., Clostridium* spp. and *Escherichia/Shigella spp*. is increased [124]. A randomized study of 23 patients with confirmed vitamin D deficiency revealed that 50,000 IU of oral vitamin D3 once weekly supplementation caused *Lactococcus* to increase, whereas *Veillonella* and *Erysipelotrichaceae* were substantially decreased after 12 weeks [113]. Vitamin D controls the expression of antimicrobial peptides, contributing to a protective effect on the intestinal and bronchial mucosa, it maintains the integrity of the mucosal barrier and it promotes epithelial healing [125].

Vitamin D develops a suppressive effect on the immune system. VDR receptors are found on the surface of dendritic cells, macrophages, T and B cells. Vitamin D inhibits the activity of the immune cells, the proliferation of B and T cells and the production of pro-inflammatory cytokines, IL-2, IL-12, IL-17, IFN-γ and TNF-α [126]. The influence of vitamin D on the development and growth of NKT cells was noted. Numerous observational studies have shown the impact of vitamin D deficiencies on the risk of autoimmune diseases such as type I diabetes, Hashimoto’s thyroiditis, inflammatory bowel disease, systemic lupus erythematosus, multiple sclerosis, vitiligo, psoriasis and rheumatoid arthritis. Supplementation of high dose vitamin D could improve the clinical course of vitiligo. Since vitamin D deficiency affects the development of autoimmune diseases, its supplementation may prevent other serious adverse events during immunotherapy of cancer [108]. 

On the other hand, vitamin D increases the production of IL-10 by DCs and cathelicidin by macrophages, and it activates Tregs as well as induces the production of IL-4 by Th2 cells together with a downregulation of the pro-inflammatory Th1, Th17 and Th9 lymphocytes. In mice, the effect of high concentrations of vitamin D on the decreased level of IL-22 was also observed. Vitamin D inhibits inflammatory cytokine production by monocytes, and suppresses dendritic cell differentiation and maturation, which helps to maintain immune tolerance [125].

### 8.5. Vitamin E

Vitamin E influences the gut microbiome and correlates with an increase in *Firmicutes* [127,128] and a significant decrease in the *Bacteroidetes cluster* [127]. In a study comparing a group with supplementation of iron and vitamin E with a group with only iron supplementation, a marked decrease in *Bacteroides* and an increase in *Firmicutes* (especially *Roseburia*) was observed. Supplementation has been positively associated with SCFA production. Thus, the addition of vitamin E to therapeutic iron supplementation may create a more favorable profile of the gut microbiome by promoting the growth of butyrate-producing bacteria [129]. Higher levels of vitamin E have also been associated with greater abundance of *Akkermansia* and other health-promoting taxa such as *Lactobacillus, Bifidobacterium* and *Faecalibacterium* [102,130]. The research results are interesting, but there are too few of them and further analysis is needed.

Vitamin E has an antioxidant effect and participates in immune regulation by inhibiting the NF-kB and STAT3 signaling pathways. It affects the proliferation of cells through the phosphoinositide 3-kinase (PI3K) pathway and the process of apoptosis [101]. It was shown that the level of vitamin E in the blood was significantly lower in patients with prostate cancer (n = 32, mean concentration—5.2 µg/mL) than in healthy subjects (n = 40, mean concentration—14.2 µg/mL) [131]. Zhang et al. reported that vitamin E intake could reduce colorectal cancer risk [132]. A study involving 278 cases of lung cancer and 205 cases of prostate cancer presented a significantly lower concentration of α-tocopherol in cancer patients than in healthy volunteers [133]. Recent studies indicated the importance of vitamin E in immunotherapy with immune checkpoints inhibitors. Cancer patients with vitamin E supplementation during ICIs treatment had a prolongation of survival. Vitamin E acts on DCs through the SCARB1 receptor and inhibits the protein tyrosine phosphatase SHP1, the internal checkpoint of dendritic cells [134]. Cross-presentation of antigen triggered systemic, antigen-specific T-cell antitumor immunity. Combining immunotherapy with vitamin E supplementation could significantly increase the effectiveness of oncological treatment.

### 8.6. Vitamin K

Vitamin K supports the diversity of bacteria in the gut microbiome. In a group of Japanese women whose diets were poor in vitamin K, a high relative abundance of *Ruminococcaceae* and *Bacteroides* was demonstrated, while in a group of women on a vitamin-K-rich diet, a high relative abundance of *Bifidobacterium* and *Lactobacillales* was found [135]. The cytotoxic and antitumor properties of vitamin K result from the reactivity of the quinone moiety of this molecule, which generates oxygen free radicals. This effect is also associated with a change in the cell cycle at the transcriptional level and a disturbance in carboxylation biochemistry [94]. Vitamin K influences cell proliferation by enhancing the expression of protein kinase A and inhibiting NF-κB by suppressing IκB kinase. The anti-tumor activity of the vitamin K analog—PPM-18 has also been demonstrated in a group of bladder cancer patients [136]. PPM-18 activated AMP-activated protein kinase (AMPK) and inhibited the PI3K/AKT and mTORC1 pathways in bladder cancer cells, inducing autophagy and apoptosis of cancer cells. There are no studies analyzing the efficacy of combination therapy with anticancer drugs and vitamin K supplementation. 

### 8.7. Iron

In vivo and in vitro studies have shown adverse effects of oral Fe supplementation on the composition of the gut microflora, gut metabolism and gut health. Changes in the bacterial composition caused by the administration of iron influenced the bacterial pathway of *Staphylococcus aureus,* suggesting a reduced protective gut microbiota response to bacterial infections and carcinogenesis [137]. Paganini and co-authors demonstrated a relationship between higher iron concentration and a decline in the amount of lactic acid bacteria—*Bifidobacterium* and *Lactobacillus* and an increase in enteropathogenic strains such as *Escherichia coli* [138]. In a group of 35 children in Kenya, after 4 months of Fe using, there was a lower number of the genus *Lactobacillus* (*p* = 0.048) and *Bifidobacterium* (*p* = 0.058), and a higher number of *Clostridiales* (*p* = 0.015) and *Enterobacteriaceae* family (*p* = 0.086), compared to the control group. Moreover, the number of *Ruminococcaceae, Lachnospiraceae* (mainly *Dorea, Blautia* and *Coprococcus*) and *Erysipelotrichaceae* was significantly higher in the group with Fe supplementation compared to the control group (*p* < 0.05 for all of the listed ones). Another study, involving 139 children aged 6–14 years, showed that after 6 months of iron supplementation, there was a significant increase in the number of *Enterobacteriaceae* (*p* < 0.005) and a decrease in the number of *Lactobacillus* (*p* < 0.0001). Moreover, an increase in concentration of fecal calprotectin (*p* < 0.01), which is a marker of enteritis, has been observed [139]. Simonyte et al. confirmed the above observations. Their study assessed the effect of various doses of supplemented iron on the microbiota and it was found that the consumption of a high dose of iron for 45 days reduced the amount of *Bifidobacterium* (*p* < 0.001, 60% vs. 78%) and *Lactobacillus* (*p* < 0.007, 8% vs. 42%), while no increase in pathogenic bacteria was observed [137]. It was also observed that an increase in SCFAs enhances iron absorption. In infants with iron deficiency, a reduction in the abundance of butyrate-producing species such as *Butyricicoccus, Roseburia* and *Coprococcus* was shown [35].

High iron level is associated with high risk of colorectal cancer by increasing the activity of bacterial enzymes involved in carcinogenesis of the large intestine. *Bifidobacteriaceae* are able to bind Fe in the large intestine and reduce the formation of free radicals. *Lactobacillus fermentum* reduces Fe (III) to Fe (II) and increases its absorption [140]. A very interesting discovery regarding the use of iron oxide nanoparticles (IONP) in cancer immunotherapy was described by Chung et al. [141]. IONPs can be used to stimulate immune cells to increase their activity at the tumor site and improve the response to cancer immunotherapy. Iron from IONPs is widely present in the organism, it is used by various cellular processes and it is biodegradable, which are advantages over other nanoparticles. IONPs can be converted to exhibit specific physicochemical properties, such as, for example, surface charge. By these modifications, the surface of the IONPs can be tailored to conjugate therapeutic agents as well as antibodies that are selective and specific for a particular cell type. The modified surface enables the delivery of antigens, adjuvants and therapeutic agents to immune cells, and the conjugation of antigens to IONPs protects them from degradation in vivo. IONPs’ labeling of immune cells, such as DCs, macrophages and lymphocytes, increases the number of cells in the vicinity of the tumor. The magnetic properties of IONPs in MRI enable the targeting of molecules to specific tumor sites. Labeling with antibodies against immune checkpoints (e.g., anti-PD-L1) could increase the effectiveness of immunotherapy. In tumor-bearing C57BL/6 mice with anti-PD-L1 antibodies on the IONPs’ surface, higher tumor growth suppression and survival rate were observed. IONPs can also be used in photothermal therapy to induce an immune response in tumors and provide immunostimulants to strengthen the effect.

### 8.8. Zinc

Dietary zinc deficiency affects the gut microbiota, and the gut microbiota affects the absorption of zinc in the intestines. Numerous studies indicate that prophylactic doses of zinc oxide (ZnO) in various animal models reduced the presence of anaerobic Gram-negative bacteria [142]. The supply of zinc caused a decrease in *Enterobacteriaceae*, mainly *Escherichia coli* and *Clostriudium* spp., in pigs. Coated ZnO nanoparticles increased the microbial richness of *Ruminococcus flavefaciens* and *Prevotella* [143,144]. Data on changes in the number of *Lactobacillus* are inconsistent. Recent studies have shown a decrease in the level of *Lactobacillus* after zinc supplementation [144,145,146]; however, there are reports of an increase in the level of this bacterium after the use of Zn [147]. In studies carried out on chicks, it was shown that in the caecum of individuals with Zn deficiency a higher number of *Proteobacteria* and a smaller number of *Firmicutes* were observed compared to the group with iron supplementation. The *Firmicutes:Proteobacteria* ratio was significantly lower in the Zn-deficient group [148]. Moreover, in the Zn-deficient group, the number of *Bacteroidetes* increased while the *Actinobacteria* decreased. Zn-deficient chicks had a significantly higher relative abundance of *Enterococcus, Enterobacteriaceae* and *Ruminococcaceae*, and a significantly lower relative abundance of *Clostridiales* and *Peptostreptococcaceae*, compared to the Zn-supplemented group.

Zinc improved the function of the immune system by increasing the level of pro-inflammatory cytokines such as IL-1β, IL-2, IL-6, TNF-α and IFN-γ and reducing IL-10 in the serum, confirming the central role of Zn for cytokine production and immunoregulation [148,149]. Zinc supplementation in combination with the probiotic *Lactobacillus plantarum IS-10506* for 90 days resulted in a significantly increased humoral immune response [147]. Evidence from several studies showed that abnormal zinc metabolism has been closely associated with the development of various types of malignancies including breast, pancreatic, lung, liver, stomach, cervical and prostate cancer [150]. Cells of prostate cancer contained 62–85% less zinc than cells from normal tissues. Low zinc level has been correlated with a higher tendency to cancer progression and the advancement of the neoplastic process. In the treatment of prostate cancer, zinc can induce a strong necrotic response by activating ERK1/2 and protein kinase C (PKC) in tumor cells [150]. Similar to the above-mentioned iron, research is underway on the use of zinc in the form of nanoclusters in combination with bovine serum albumin (ZnS@BSA) [151]. Zinc ions are released in the tumor’s acid microenvironment and activate the signaling pathway of cyclic guanosine monophosphate-adenosine monophosphate synthase/stimulator of interferon genes (cGAS/STING). This causes an influx of CD8+ T cells to the tumor and dendritic cells, leading to improved immunotherapy efficacy, what has been proven in mice with hepatocellular carcinoma.

### 8.9. Magnesium

Little research has been done on the effects of magnesium on the gut microbiome. It has been shown so far that low doses of magnesium in the diet (30 mg/kg) resulted in a greater abundance of selected bacterial species in mice and a higher presence of rare species than in a high magnesium supplementation diet (4000 mg/kg) [152]. The intestinal microbiota of hypo-Mg mice showed a lower relative abundance of *Actinomycetes* and *Proteobacteria* while a higher abundance *of Bacteroidetes, Clostridiales* and *Clostridiaceae.* In hyper-Mg mice, a higher abundance of *Bifidobacterium, Adlercreutzia* and *Lachnospiraceae* was observed. In studies on liver damage caused by methotrexate therapy, it was shown that magnesium Isoglycyrrhizinate (MgIG) changed the composition of the intestinal microbes by increasing the level of probiotic *Lactobacillus* and reducing the level of *Muribaculaceae* [153]. Another study demonstrated that mice on a low-magnesium diet showed a lowered number of *Bifidobacterium* in the gut and high levels of TNF-α and IL-6 [154].

**Table 2 cancers-14-05577-t002:** The relationship between micronutrients and microbiome.

Micronutrient	Influence on Microbiome	Reference
Vitamin A	*Akkermansia* ↑	[83]
	*Bifidobacterium* ↑	[83,85]
	*Proteobacteria* to *Actinobacteria* ratio ↑ *Proteobacteria* to *Firmicutes* ratio ↑	[84]
	*Bacteroidetes* ↑ *Proteobacteria* ↑ *Actinobacter* ↑ *Enterobacter* ↑	[85]
Vitamin B1	*Bacteroides* ↑, *Faecalibacterium* ↑, *Prevotella* ↑	[99]
Vitamin B2	*Faecalibacterium* ↑, *Roseburia* ↑, *Enterobacteriaceae* ↓	[101,102]
	*Alistipes shahii* ↑	[103]
Vitamin B3	*Bacteroidetes* ↑	[103]
Vitamin B12	*Akkermansia* ↑	[107]
Vitamin C	*Collinsella* ↑	[103]
	*Bacteroides* ↑, *Roseburia* ↑, *Faecalibacterium* ↑, *Akkermansia* ↑, *Bifidobacterium* ↑	[109]
Vitamin D	*Lactobacillus rhamnosus* ↑, *Lactobacillus plantarum* ↑	[115]
	*Clostridium difficile* ↓	[114,116]
	*Akkermansia* ↑, *Faecalibacterium* ↑, *Coprococcus* ↑	[117]
	*Enterobacteria* ↑	[118]
	*Bifidobacterium longum* ↑	[119,120]
	*Ruminococcus* ↑, *Mogibacterium* ↑, *Blautia* ↑	[121]
	*Alistipes* ↑, *Roseburia* ↑, *Parabacteroides* ↑	[122,123]
	*Pediococcus* ↑, *Clostridium* ↑, *Escherichia* ↑, *Shigella* ↑	[124]
Vitamin E	*Firmicutes* ↑	[127,128]
	*Bacteroidetes cluster* ↓	[127]
	*Akkermansia* ↑, *Lactobacillus* ↑, *Bifidobacterium* ↑, *Faecalibacterium* ↑	[102,130]
Vitamin K	*Bifidobacterium* ↑, *Lactobacillales* ↑	[135]
Iron	*Bifidobacterium* ↓, *Escherichia coli* ↑, *Ruminococcaceae* ↑, *Lachnospiraceae* ↑, *Erysipelotrichaceae* ↑	[138]
	*Enterobacteriaceae* ↑	[139]
	*Lactobacillus* ↓	[138,139]
Zinc	*Escherichia coli* ↓, *Clostriudium* ↓	[142]
	*Ruminococcus flavefaciens* ↑, *Prevotella* ↑	[143,144]
	*Lactobacillus* ↓	[144,145,146]
Magnesium	*Bifidobacterium* ↑, *Adlercreutzia* ↑, *Lachnospiraceae* ↑	[152]
	*Lactobacillus* ↑, *Muribaculaceae* ↓	[153]
Selenium	*Lachnospiraceae* ↑, *Ruminococcaceae* ↑, *Christensenellaceae* ↑, *Lactobacillus* ↑	[155]

↑-increase in the number of bacteria; ↓-decrease in the number of bacteria.

### 8.10. Selenium

Selenium supplementation determined an increase in the number of families *Lachnospiraceae, Ruminococcaceae*, *Christensenellaceae* and *Lactobacillus* in the gut microbiome [155]. Selenium stimulates the functionality of NK cells, macrophages and neutrophils as well as the production of IFN-γ, TNF-α and IL-6 [155]. Selenium supplementation enhances the cytotoxic functions of NK cells in mice by increasing the expression of IL-2 receptors (IL-2R) on their surface. Consumption of Se-enriched foods (200 mg per serving) over 3 days increased the levels of IL-2, IL-4, IL-5, IL-13 and IL-22, indicating that selenium promotes the immune response of Th1 and Th2 cells. Selenium is anti-cancerous in the tumor microenvironment, inhibiting the proliferation of cancer cells. In leukemia, sodium selenite (200 mg/day for 8 weeks) increased the CD8+ T cells-mediated tumor cytotoxicity and NK cell activity. Similar observations have been made in different neoplasms [156]. It strengthens the immune system by regulating the production of antibodies. Immunomodulating functions of selenium could reduce immunorelated adverse events (irAEs) of cancer immunotherapy [157].

### 8.11. Omega-3

Studies comparing the effects of different doses of omega-3 polyunsaturated fatty acids (PUFAs) showed that supplementation with low doses of PUFAs (30 mg) was associated with a higher concentration of *Bacteroidales, Clostridium, Eubacterium* and *Planococcaceae* and with a lower abundance of *Lactobacillus, Helicobacter* and *Ruminococcus* than supplementation with 60 mg or 90 mg of the omega-3 PUFAs [158]. *Firmicutes, Clostridiales, Lactobacillus* and *Bifidobacterium* were relatively more abundant in the group with supplementation of 60 mg of PUFAs, while *Bacteroidetes* were less abundant in this group. However, in the group treated with 90 mg omega-3 PUFAs, there was a much higher concentration of *Helicobacter, Jeotgalicoccus, Staphylococcus, Ruminococcus* and *Alcaligenaceae* than in remaining groups. A higher percentage of *Lactobacillus, Helicobacter* and *Ruminococcus* and a lower percentage of *Bacteroides, Clostridium* and *Prevotella* was observed in groups receiving higher dose of PUFAs (60 and 90 mg). Studies on mice showed that omega-3 supplementation resulted in enrichment of the intestinal microbiome in *Bifidobacterium spp., Bacteroidetes, Lactobacillus spp. (Firmicutes), Enterobacteriales (Proteobacteria), Lactococcus, Eubacterium, Lachnospermaceae, Ruminococcansiaae* and *Akkermansia* [159].

The effects of omega-3 PUFAs on the immune system is manifested by the inhibition of release of pro-inflammatory cytokines: tumor necrosis factor-α (TNF-α), IL-1, IL-6, IL-8 and IL-12, as well as by activation of nuclear factor kappa B [159]. In addition, omega-3 PUFAs modulate the activity of immune cells, primarily neutrophil function, including migration and phagocytic capacity, as well as the production of reactive oxygen species and cytokines. It stimulates macrophages to produce and secrete cytokines and chemokines, and increases the ability to phagocytose. Moreover, PUFAs modulate the activation of T cells.

## 9. Conclusions

The effectiveness of immunotherapy depends on the activity of the host’s immune system and its ability to defeat tumor escape mechanisms from immune surveillance. The transformation of “cold tumors” into “hot tumors” is associated with an increased number of immune cells and cytokines in the tumor microenvironment, an increased tumor cells apoptosis and a greater effectiveness of immunotherapy. The intestinal microbiome is a proven immune system modulating factor, which positive role has been observed in numerous cancers. An increase in the level of Th CD4+ and CD8+ cells and a decrease in Tregs in the tumor microenvironment correlate with the abundance of *Bifidobacterium, Faecalibacterium* genus *Ruminococcaceae* family and *Clostridiales* order in the gut. Increased TNF-α production is correlated with greater abundance of *Bacteroidetes*, especially *Alistipes*. A higher percentage of peripheral blood NK cells and memory CD8+ T cells have been associated with higher numbers of *Alistipes putredinis, Prevotella copri* and *Bifidobacterium longum*. Richness of *Bifidobacterium spp*., *Akkermanisa muciniphila* and *Enterococcus hire* improves anti-tumor specific immunity and the response to anti-PD-1 or anti-PD-L1 immunotherapy by activating antigen-presenting cells and cytotoxic T cells within the tumor. Moreover, *Bifidobacterium spp. Bacteroides*, especially *B. fragilis* and *B. thetaiotamicron*, may affect the IL-12- and IFN-γ-dependent Th1 immune response and enhance the efficacy of immunotherapy.

Micronutrients affect directly the activities of the immune system or regulate their function by influencing the composition of the microbiome. Therefore, micronutrients can significantly influence the effectiveness of immunotherapy and the development of irEAs. This may become an interesting direction for further research on the predictors of immunotherapy.

Vitamin A supplementation may contribute to the growth of *Bifidobacterium, Akkermansia*, *Proteobacteria* and *Firmicutes*, the richness of which have been correlated with the greater effectiveness of immunotherapy in NSCLC, RCC, hepatocellular and colorectal cancer patients. Retinol is required for B-cell stimulation, thymocyte growth and NK cell activation, leading to inhibition of tumor cell proliferation and tumor growth arrest. B vitamins affect the increased level of TNF-α and IL-6 in the tumor microenvironment, the activation of NF-κB factor in macrophages and the increased abundance of *Bacteroides, Faecalibacterium* and *Prevotella*, which stimulate the response to immunotherapy in NSCLC and melanoma patients. Vitamin B2 stimulates T cell activation and enriches the overall microbiome diversity, especially in *Alistipes*, which may increase the effectiveness of NSCLC, RCC, sarcoma and melanoma immunotherapy. Vitamin B12 particularly stimulates growth of *Akkermansia muciniphila* and *Eubacterium halli* and the immune response mediated by NK and T cells. Vitamin C supplementation enriches the microbiome in *Roseburia, Faecalibacterium, Akkermansia, Bifidobacterium* and *Bacteroides* and may increase the effectiveness of the treatment of numerous neoplastic diseases, including leukemia, colorectal cancer, melanoma, pancreatic cancer, prostate cancer, NSCLC, breast cancer and ovarian cancer. Vitamin C increases the immunogenicity of effector T cells in murine models and inhibits tumor growth. Vitamin D and E stimulate the growth of *Lactobacillus, Akkermansia, Faecalibacterium, Firmicutes, Actinobacteria, Proteobacteria* and *Bifidobacterium*, which enhance the effectiveness of immunotherapy in melanoma patients. Zinc enriches the microbiome in *Prevotella, Proteobacteria* and *Actinobacteria* and improves the function of the immune system by increasing the level of pro-inflammatory cytokines such as IL-1β, IL-2, IL-6, TNF-α and IFN-γ in the tumor environment. Moreover, zinc has been studied as a drug transporter directly to the tumor. Magnesium promotes the growth of *Bifidobacterium* in the intestines and increases the level of TNF-α and IL-6. Selenium enriches the microbiome in *Lachnospiraceae, Ruminococcaceae, Christensenellaceae* and *Lactobacillus*, which may favor the therapy in hepatocellular carcinoma patients.

## Figures and Tables

**Table 1 cancers-14-05577-t001:** Gut microbiota species associated with efficacy of cancer immunotherapy.

Malignancy	Treatment	Bacteria Correlated with Positive Immunotherapy Response	Reference
Non-small-cell lung cancer	anti-PD-L1	*Akkermansia muciniphila, Alistipes, Enterococcus hirae*	[49,50,71]
		*Bifidobacterium bifidum*	[50,72]
		*Alistipes putredinis, Prevotella copri, Bifidobacterium longum*	[71]
		*Methanobrevibacter and Parabacteroides*	[70]
		*Bifidobacterium breve, Propionibacterium acnes, Prevotella copri, Rikenellaceae, Staphylococcus aureus, Streptococcus, Peptostreptococcus, Oscillospira, Faecalibacterium prausnitzi, Bacteroides plebeius, Enterococcus hirae, Enterobacteriaceae*	[7,71,73,74]
Melanoma	anti-PD-L1	*Bifidobacterium, Akkermansia muciniphila, Alistipes, Enterococcus hirae, Faecalibacterium prausnitzii, Bacteroides thetaiotamicron, Holdemania filiformis, Bacteroides caccae*	[62]
	anti-PD-L1, anti-CTLA-4	*Ruthenibacterium lactatiformans, Eubacterium limosum, Fusobacterium ulcerans, Phascolarctobacterium succinatutens, Bacteroides uniformis, Bacteroides dorei, Paraprevotella xylaniphila, Parabacteroides distasonis, Parabacteroides johnsonii*	[56,68]
		*Enterococcus faecium, Collinsella aerofaciens, Bifidobacterium adolescentis, Klebsiella pneumoniae, Veillonella parvula, Parabacteroides merdae, Lactobacillus spp., Bifidobacterium longum*	[69]
		*Bifidobacterium longum, Collinsella aerofaciens, Enterococcus faecium*	[49,61]
Sarcoma	anti-PD1 or anti-CTLA-4	*Bacteroides fragilis, Bacteroides thetaiotamicron, Burkholderia,* *Akkermansia muciniphila, Enterococcus hirae, Alistipes*	[49,52]
Colorectal cancer	anti-PD1 or anti-CTLA-4	*Ruthenibacterium lactatiformans, Eubacterium limosum, Fusobacterium ulcerans, Phascolarctobacterium succinatutens, Bacteroides uniformis, Bacteroides dorei, Paraprevotella xylaniphila, Parabacteroides johnsonii, Parabacteroides gordonii, Alistipes senegalensis*	[68]
Renal cell carcinoma	anti-PD-L1	*Akkermansia muciniphila, Lachnospiraceae, Erisypelotrichaceae lacteria, Enterococus faevium, Alistipes indistinctus, Bacteroidaceae, Bacteriodes xylanisolvens, Bacteroides nordii*	[49]
Carcinoma hepatocellulare	anti-PD-1	*Streptococcus thermophilus, Fusobacterium ulcerans, Candidatus Liberibacter, Lactobacillus mucosae, Ruminococcus obeum, unclassified Lachnospiracae, Ruminococcus bromii, Subdoligranulum, Bacteroides cellulosyticus, Lactobacillus gasseri, Anaerotruncus colihominis, Eubacterium hallii, Dorea formicigenerans, Lactobacillus vaginalis, Dalister invisus, Lactobacillus oris, Akkermansia muciniphila, Bifidobacterium dentium, Megasphera micronuciformis, Coproccus comes*	[75]

*Abbreviations*: PD-1—programmed death-1, PD-L1—programmed death-1 ligand, CTLA4—cytotoxic T cell antigen 4.

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
