# Peer review of "Interactions between Dietary Micronutrients, Composition of the Microbiome and Efficacy of Immunotherapy in Cancer Patients"

_cancers, 2022, doi:10.3390/cancers14225577_

Round 1

Reviewer 1 Report

This review presents a wide spectrum of factors that affect immunotherapy in cancer. Generally, authors’ approach is comprehensive and up-to date but the manuscript needs a revision:

·         The title of section 3 is “The role of Short-Chain Fatty Acids”, but the authors discuss the role of fiber. The title should be changed and the more comprehensive information about and the role of short-chain fatty acids should be added

·         Table 1 - please match the references with information in all columns of the table

·         In section 7 the same information is repeated as in section 3.

·         The title of section 7 is “Micronutrients, gut microbiome and cancer” but again you discuss also probiotics, prebiotics and omega-3 acids. The title should be changed.

·         In section 7 you can add a table summarizing the most important information on the relationship of microelements with microbiota and cancer. Then you do not need to describe again their role in the conclusions. This will improve the clarity of the entire chapters.

·         There is lack of references in paragraphs: 116, 149-161, 198, 257, 295, 301, 476-488, 535, 825

·         Some articles concerning the topic were recently published and should be cited.

The examples are:

Prebiotics, Probiotics, and Postbiotics in the Prevention and Treatment of Anemia.

Zakrzewska Z, Zawartka A, Schab M, Martyniak A, SkoczeÅ„ S, Tomasik PJ, WÄ™drychowicz A. Microorganisms. 2022 Jun 30;10(7):1330. doi: 10.3390/microorganisms10071330. PMID: 35889049

The Role of Nutritional Status, Gastrointestinal Peptides, and Endocannabinoids in the Prognosis and Treatment of Children with Cancer.

Schab M, Skoczen S. Int J Mol Sci. 2022 May 5;23(9):5159. doi: 10.3390/ijms23095159. PMID: 35563548

Author Response

Reviewer 1

  1. The title of section 3 is “The role of Short-Chain Fatty Acids”, but the authors discuss the role of fiber. The title should be changed and the more comprehensive information about and the role of short-chain fatty acids should be added
  2. Table 1 - please match the references with information in all columns of the table
  3. In section 7 the same information is repeated as in section 3.
  4. The title of section 7 is “Micronutrients, gut microbiome and cancer” but again you discuss also probiotics, prebiotics and omega-3 acids. The title should be changed.
  5. In section 7 you can add a table summarizing the most important information on the relationship of microelements with microbiota and cancer. Then you do not need to describe again their role in the conclusions. This will improve the clarity of the entire chapters.
  6. There is lack of references in paragraphs: 116, 149-161, 198, 257, 295, 301, 476-488, 535, 825
  7. Some articles concerning the topic were recently published and should be cited.

The examples are:

Prebiotics, Probiotics, and Postbiotics in the Prevention and Treatment of Anemia.

Zakrzewska Z, Zawartka A, Schab M, Martyniak A, SkoczeÅ„ S, Tomasik PJ, WÄ™drychowicz A. Microorganisms. 2022 Jun 30;10(7):1330. doi: 10.3390/microorganisms10071330. PMID: 35889049

The Role of Nutritional Status, Gastrointestinal Peptides, and Endocannabinoids in the Prognosis and Treatment of Children with Cancer.

Schab M, Skoczen S. Int J Mol Sci. 2022 May 5;23(9):5159. doi: 10.3390/ijms23095159. PMID: 35563548

Response : We are very grateful for Your comment. Your opinion is very valuable to us and contributes to even greater diligence in our work.

Point 1. The title of section 3 has been changed to: “The role of Dietary Fiber and Short-Chain Fatty Acids” and we have added more information about short-chain fatty acids

Point 2. The table has been improved and references has been matched

Point 3. The repeated information have been removed.

Point 4. The title of the section 7 has been changed to “Probiotics and Prebiotics” and the section 8 “Micronutrients, gut microbiome and cancerhas been added, to make information more comprehensive.

Point 5. The table which includes summarize of correlation between micronutrients and microbiome has been added.

Point 6. All missing references has been added.

Point 7. Indicated articles has been added to references in point 35 and 82.

Reviewer 2 Report

A very interesting article. I hope that other scientists will also appreciate it :) below are some comments

please standardize the font and size in the article (in accordance with the magazine's guidelines)

line 52: I can't see the dot after [4,5,6]

line 338: "or" without italics

Table 1 - Please explain the abbreviations below the table

line 445: why are you writing about unpublished data?

line 525: "... group received placebo. [88]." remove the unnecessary period

line 592: useless space

line 598 "plantarum in italics

line 764: "... (p 0.007.8% vs 42%)," missing <or = (I don't know)

line 811: "IS-10506" in italics

References

when quoting articles - the year of publication should be bold

I'm not sure, but some references are in black and some in gray. Please standardize it.

Author Response

A very interesting article. I hope that other scientists will also appreciate it :) below are some comments.

  1. please standardize the font and size in the article (in accordance with the magazine's guidelines)
  2. line 52: I can't see the dot after [4,5,6]
  3. line 338: "or" without italics
  4. Table 1 - Please explain the abbreviations below the table
  5. line 445: why are you writing about unpublished data?
  6. line 525: "... group received placebo. [88]." remove the unnecessary period
  7. line 592: useless space
  8. line 598 "plantarum in italics
  9. line 764: "... (p 0.007.8% vs 42%)," missing <or = (I don't know)
  10. line 811: "IS-10506" in italics
  11. References when quoting articles - the year of publication should be bold
  12. I'm not sure, but some references are in black and some in gray. Please standardize it.

Response We are very grateful for Your comment. Your opinion is very valuable to us and contributes to even greater diligence in our work.

Point 1. The font and size in the article have beed standardized to “Calibri”, size 10.

Point 2. The dot has been added.

Point 3. The italics has been removed.

Point 4. Abbreviations have beed added.

Point 5. Mentioned data is our own work based on our investigation in cancer patients, which hasn’t been published yet. We included it in our paper because we believe that it well complement topic of microbiome and cancer immunotherapy.

Point 6. Period has been removed.

Point 7. Space has been removed

Point 8. Italics has been removed

Point 9. The “<” has been added, so the final sentence is “p<0.007.8% vs 42%)"

Point 10. Italics has been removed.

Point 11. The year of publications has been bold in references.

Point 12. The color and font of the references have been standardized.

Reviewer 3 Report

In the current review manuscript, titled as "Interactions between dietary micronutrients, composition of the microbiome and efficacy of immunotherapy in cancer patients", the authors aimed to describe the relationship between the sypply of microelements, gut microbiota and immunotherapy effectiveness. Also to describe immune system changes upon dysbiosis and changes in microelements supply.

Overall, the manuscript is very well written with sufficient information. Only some minor changes are needed.

1. The authors should revise the manuscript for the english language, as some minor errors have been detected.

2. Since there is too much information, additional tables and/or figures would help understanding  of the points.

Author Response

Response We are very grateful for Your comment. Your opinion is very valuable to us and contributes to even greater diligence in our work.

Point 1. In line 62, “furthermore” has been added in the beginning of the sentence.

In line 78 “It’s has been changed to “Its”.

In line 417 “abundace” has been changed to “abundance”.

In line 761 “sp” has been changed to “spp”.

In line 869 “immunoteharpy” has been changed to “immunotherapy”.

Point 2. The table 2 ” The relationship between micronutrients and microbiome” has been added in section 8.

Round 2

Reviewer 1 Report

The authors responded properly to the revision.

Reviewer 2 Report

-